# Direct and Indirect Bactericidal Effects of Cold Atmospheric-Pressure Microplasma and Plasma Jet

**DOI:** 10.3390/molecules26092523

**Published:** 2021-04-26

**Authors:** Ahmad Guji Yahaya, Tomohiro Okuyama, Jaroslav Kristof, Marius Gabriel Blajan, Kazuo Shimizu

**Affiliations:** 1Graduate School of Science and Technology, Shizuoka University, Hamamatsu 832-8561, Japan; shimizu@cjr.shizuoka.ac.jp; 2Graduate School of Integrated Science and Technology, Shizuoka University, Hamamatsu 432-8561, Japan; Okuyama.tomohiro.16@shizuoka.ac.jp; 3Organization for Innovation and Social Collaboration, Shizuoka University, Hamamatsu 432-8561, Japan; jaroslav.kristof@gmail.com (J.K.); blajanmarius@yahoo.com (M.G.B.)

**Keywords:** DBD microplasma, plasma jet, sterilization, plasma activated water, UV-Vis spectroscopy, reactive oxygen, nitrogen species

## Abstract

The direct and indirect bactericidal effects of dielectric barrier discharge (DBD) cold atmospheric-pressure microplasma in an air and plasma jet generated in an argon-oxygen gas mixture was investigated on *Staphylococcus aureus* and *Cutibacterium acnes*. An AC power supply was used to generate plasma at relatively low discharge voltages (0.9–2.4 kV) and frequency (27–30 kHz). Cultured bacteria were cultivated at a serial dilution of 10^−5^, then exposed to direct microplasma treatment and indirect treatment through plasma-activated water (PAW). The obtained results revealed that these methods of bacterial inactivation showed a 2 and 1 log reduction in the number of survived CFU/mL with direct treatment being the most effective means of treatment at just 3 min using air. UV–Vis spectroscopy confirmed that an increase in treatment time at 1.2% O_2_, 98.8% Ar caused a decrease in O_2_ concentration in the water as well as a decrease in absorbance of the peaks at 210 nm, which are attributed NO_2_^−^ and NO_3_^−^ concentration in the water, termed denitratification and denitritification in the treated water, respectively.

## 1. Introduction

Conventional methods for sterilization such as wet/dry heat, chemical gases, or irradiation have several disadvantages [1]. These include negatively altering the physical and biological performance of material properties such as molecular weight, morphology, and volume of sensitive materials, including polymeric biomaterials [2,3,4,5]. Atmospheric plasmas were first used for sterilization of contaminated matter by [6]. This technology led to the development of many plasma devices for microbial inactivation on solid surfaces [7]. Cold atmospheric plasma (CAP) technology is a promising tool for surface decontamination and hand disinfection in public health and hospital care, which does not have the above-mentioned disadvantages [8,9,10,11]. CAP is a weakly ionized gas that can operate under atmospheric conditions near or just above room temperature [12]. Most plasmas are created when extra energy is added to a gas, knocking electrons free from atoms. Only a small fraction of gaseous atoms and molecules, which are the main carriers of heat, collide with the highly energetic free electrons resulting in further excitation, ionization, and dissociation, while the plasma remains cold [11]. These properties permit the disinfection or sterilization of thermosensitive materials and allow in vivo applications, opening a new and larger spectrum of possible applications [7,8,13,14,15,16,17,18]. Microplasma is a type of dielectric barrier discharge (DBD) atmospheric-pressure plasma that can be generated at a relatively low discharge voltage of about 1 kV. The micrometer-order discharge gap allows microplasma to be generated at atmospheric pressure and voltages from 0.4 kV depending on the discharge parameters [19].

Plasma jets also belong to the large family of cold atmospheric pressure plasmas; they are discharged plasma that can extend beyond the plasma generation region into the surrounding ambient environments, and they could be generated at a discharge voltage around 2 kV, with discharge gaps in the order of millimeters [19]

Multiple research results [20,21,22,23,24] have revealed that the reactive species generated by the plasma play the most important role in the plasma sterilization process. Multiple kinds of active species are considered to enhance the antimicrobial effects of plasma in general; they include various types of electrons, ions, radicals, UV light, electric fields, and metastable [25]. Depending on plasma generation setup, humidity, pressure, and gas used, highly reactive oxygen-nitrogen species (RONS; O, O_2_^−^ O_3_, OH, OOH, and H_2_O_2_; and NO, NO_2_, NO_3_, N_2_O_3_, N_2_O_4_, and ONOO^–^) in a liquid or gas phase can be generated, and these species can efficiently inactivate micro-organisms by damaging their nucleic acids, lipids, and proteins [26,27,28,29].

This research investigated the low discharge voltage bactericidal potential of direct microplasma treatment in comparison with indirect plasma jet treatment through plasma-activated water (PAW) using the most abundant gas (air, argon, and oxygen) on *Staphylococcus aureus* and *Cutibacterium acnes*. We also investigated some properties of PAW such as the effects of increasing O_2_ concentration in the working gas mixture. In addition, the effect of increasing the treatment time on the concentration of O_2_ and RONS in water and the effect of O_3_ concentration on denitrification were also discussed.

The direct plasma jet’s area of bacterial inactivation is narrow, and completely depends on the electrode’s cross-sectional area. To improve that, PAW was used and termed “indirect treatment”. A low discharge voltage is advantageous since it means a smaller power supply, less electrical insulation, and portable devices. Multiple papers [30,31,32,33] demonstrated direct and indirect plasma sterilization possibilities at high discharge voltages up to 40 kV through PAW. This can be lowered by altering the plasma discharge parameters such as frequency, waveform, gas, and discharge gap.

*S. aureus* is a major human pathogenic bacterium that causes a wide variety of clinical manifestations [34]. These types of infections are commonly acquired in both community and hospital settings, and treatment remains challenging in the management due to the emergence of multi-drug resistant strains such as MRSA (methicillin-resistant *Staphylococcus aureus*) [35,36]. *Propionibacterium acnes* is a facultative anaerobic Gram-positive rod, abundant on the human skin, and mainly associated with the sebaceous glands of the shoulder and axilla [37]. It is most commonly associated with the chronic skin disease *acne vulgaris*. However, it may also cause bone and joint infections, including implant-associated infections. *P. acnes* has been recognized as an emerging cause of shoulder infections [38,39].

## 2. Results

The bactericidal properties of direct (microplasma) treatment on *C. acne* and *S. aureus* using air and indirect treatment through PAW on *S. aureus* using an argon-oxygen mixture as working gases at multiple mixtures by percentage, power supply, and frequencies were studied. For both methods of treatments, the bacterial concentration was a serial dilution at 10^−5^. The following exclusions were made: The effect of temperature on the bacterial inactivation process for both microplasma and plasma jet were neglected. This was because the highest electrode temperature recorded for both plasmas was less than 30 °C during the entire treatment process. The bactericidal effect of the electric field was also not considered, even though the bacterial fluid was not allowed to completely dry off before microplasma treatment and the distance between the electrode and samples was maintained at 2 mm. In this situation, the main functional species of atmospheric pressure plasma sterilization were the chemical substances produced by the plasma [20,21,22,23,24]. After 48 h of incubation, the mannitol salt agar changed color from red to yellow; this was because of sugar fermentation caused by *S. aureus*.

### 2.1. Direct Treatment

Figure 1a shows the results for direct microplasma treatment on *S. aureus* and Figure 1b shows the results for *C. acne*. These images were taken against a black background 48 and 72 h, respectively, after treatment. So, the dark portions on the images are regions of no bacterial growth and little changes in coloration were from different light exposures. The samples exposed to direct microplasma treatment at an AC frequency of 27 kHz with discharge voltage of 900 V using synthetic air at 2 L/min from a cylinder showed a 2-log reduction, which is equivalent to 99% reduction in CFU/mL (see Figure 1d (left)). The concentration of (+ and −) ions in the post-discharge region, 3 cm below the grounded grid, during the direct microplasma treatment discharge at 900 V were 1.9 × 10^7^ ion·cm^−3^ and 6.6 × 10^6^ ion·cm^−3^, respectively. Positive ions are generally N_2_^+^, while negative ions are O_2_^−^; these lifetimes are relatively short, but can reach the target surface [40]. The concentration of +/− ions in the microplasma post-discharge region may also depend on their reactions with external particles in the atmosphere.

### 2.2. Indirect Treatment

The bactericidal effects of PAW on *S. aureus* are shown in Figure 1c. These samples were treated at a discharge voltage of 2.4 kV for 1.2% O_2_ at an AC frequency of 30 kHz. The treatment time for PAW was 15, 30, and 45 min at a distance of 2 mm from the plasma source. These images were taken 48 h after treatment and incubation at 37 °C. To determine the change in properties of the treated water after exposure to plasma jet, the pH and temperature at different conditions were checked. The pH level slightly decreased to 6 from neutral and the temperature decreased from 25.1 to 22.7 °C. The PAW treatment showed a bactericidal effect of 1 log reduction at 45 min, which is a 90% reduction in number of colonies survived (see Figure 1d (right)).

### 2.3. Ozone Generated by Direct and Indirect Treatment

The ozone level recorded during the direct microplasma treatment in the air using an AC power supply at a discharge voltage of 900 V (0-peak) and a frequency of 27 kHz was 47 ppm. For indirect treatment using a plasma jet, the ozone generated was lower—15.5 ppm at 0.8% O_2_ and 27.1 ppm at 1.2% O_2_.

### 2.4. Power and Streamer Discharge for Direct and Indirect Treatment

The discharge voltage for 0.8% O_2_ was 2.2 kV and 1.2% O_2_ was 2.4 kV; both had the same frequency of 30 kHz. Figure 2b shows the discharge voltage and corresponding discharge current waveform with the width of the capacitive current waveform’s peak at 10 μs for a treatment time of 15, 30, and 45 min. From these values, we also calculated that the actual active time of the plasma jet streamers irradiation in 900, 1800, and 2700 s was 270, 540, and 810 s, respectively.

Discharge power was calculated using the charge–voltage curve (Lissajous curve) for high-frequency AC voltage. The applied voltage was measured with a Tektronix high-voltage probe (voltage ratio 1000:1), and the transferred charge was measured from the ground ceramic capacitor (100 nF), which was connected in series with the microplasma and plasma jet electrode. Figure 2c,d shows the Lissajours figure for microplasma and plasma jet discharge with the obtained optimum power of 3.31 W at 900 V, 27 kHz, and air working gas; and 0.56 W at 2.4 kV, 30 kHz, and 1.2% O_2_ working gas, respectively.

A streamer discharge is a type of transient electrical discharge that forms at the surface of a conductive electrode carrying a high voltage in an insulating medium such as air. From the data obtained from an oscilloscope (see Figure 2a), the approximate width of the capacitive current waveform’s peak from its base was 10 μs per circle (two peaks) at a frequency of 27 kHz and treatment time of 1, 2, and 3 min. From these values, we calculated that the actual active time of the microplasma streamers irradiation in 60, 120, and 180 s was 16.2, 32.4, and 48.6 s, respectively.

### 2.5. UV-Vis Spectroscopy of Treated Water

Long-lived RONS (H_2_O_2_, NO_2_^−^, and NO_3_^−^) in liquid phase are the main species contributing to the UV absorbance between 190 and 340 nm [41]. To confirm this result, the spectral line references by [41] of H_2_O_2_, NO_2_^−^, and NO_3_^−^ were obtained from standard solutions of H_2_O_2_, NaNO_2_, and NaNO_3_, respectively, under different concentrations. Each standard solution has a specific maximum absorption wavelength. H_2_O_2_ may be below 190 nm and those of NaNO_2_ and NaNO_3_ at about 210 nm [42]. Figure 3 (left) and (right) shows the spectral lines obtained from PAW with 0.8% O_2_ with 99.2% Ar and 1.2% O_2_ with 98.8% Ar treatment, respectively. The spectra of H_2_O_2_ and OH^-^ at 0.8% O_2_ were below 190 nm and those of NO_2_^−^ and NO_3_^−^ were at 210 nm. A negative part of the spectrum at 190 nm was also observed to decrease with an increase in treatment time from 15 to 45 min; this was attributed to the decrease in oxygen concentration in the water.

Oxygen is naturally present inside water; after Ar-O_2_ plasma treatment, part of the O_2_ is replaced by Ar. Absorbance of the samples measured were compared with a reference sample (distilled water, containing some oxygen). If the amount of oxygen in the analyzed samples was lower than in the reference samples, it resulted in negative O_2_ absorbance. A similar effect was observed by [42,43]. An increase in treatment time caused decrease in O_2_ concentration in the water as well as a decrease in absorbance of the peaks at 210 nm, which are attributed NO_2_^−^ and NO_3_^−^ concentration in the water termed denitratification and denitritification in the treated water, respectively.

The effect of dissolved oxygen concentration on the rates of denitratification and denitritification were also investigated in [44,45], which reported similar findings.

### 2.6. Effect of Ozone on RONS

The concentration of ozone almost doubled in the indirect treatment using a plasma jet from 15.5 ppm at 0.8% O_2_ to 27.1 ppm at 1.2% O_2_. The absorbance of the peaks at 210 nm, which are attributed NO_2_^−^ and NO_3_^−^ concentration, decreased with an increase in O_3_ concentration; this could be attributed to the oxidation of NO_2_^−^ by O_3_. Similar findings were reported by [46,47].

## 3. Materials and Methods

### 3.1. Direct Treatment

Dielectric barrier discharge (DBD) atmospheric microplasma [19], [22,23] (see Figure 4a) was generated at a discharge voltage of 900 V using an alternating current (AC) power supply at a frequency of 27 kHz. The electrode was gold plated with a diameter of 30 mm, a back layer of 50 µm, and a gap distance (interval between HV and ground) of 25 µm. The electrode’s configuration was a grid-shaped gold wire with an interval of 3.5 mm (see Figure 4b). The wire width was 1.2 mm on the high voltage side and 0.2 mm on the ground side. The electrode had 9 perforated holes of 2 mm between the wires, allowing the passage of gas that could transport microplasma-induced active species to the target. The air flow was set at 2 L/min by a flow meter (Yamato, Tokyo, Japan) and the treatment times were 1, 2, and 3 min. The distance between the electrode and samples on the surface of the petri dish was set at 2 mm via a grounded micrometric sample holder. The temperature of the electrode was confirmed using a thermal camera (TVS-200, Avio, Yokohama, Japan). Ions and ozone generated were measured with an ion counter (NKMH-103, Hokuto, Tokyo, Japan) and ozone monitor (EG-2001, Ebara Jitsugyo, Kanagawa, Japan), respectively.

### 3.2. Indirect Treatment

An atmospheric pressure plasma jet (see Figure 4b) used for PAW treatment was generated in an airtight chamber at a discharge voltage of 2.2 kV for 0.8% O_2_ and 2.4 kV for 1.2% O_2_, both at an AC frequency of 30 kHz using a Neon transformer (M-5, Alpha Neon, Gifu, Japan). The treatment times for PAW were 15, 30, and 45 min at a distance of 2 mm from the plasma source. PAW samples treated at 2.4 kV, 1.2% O_2_ at an AC frequency of 30 kHz for 15, 30, and 45 min at a distance of 2 mm from the plasma source were used for *S. aureus* inactivation.

The setup was constructed from a needle-type powered (tungsten steel) electrode with a diameter and length of 0.8 and 110 mm, respectively. The electrode was enclosed in a quartz tube with an inner diameter of 2.5 mm, outer diameter of 4.3 mm, and length of 110 mm. The distance between the electrode and petri dish containing 5 mL of distilled water was set to 2 mm via a micrometric sample holder attached to the chamber. The flow of the gas mixture was maintained at 3 L/min by a flow meter containing 98.8% Ar and 1.2% O_2_, and 99.2% Ar and 0.8% O_2_ measured by an oxygen monitor (OXY-1, Jikco, Osaka, Japan). This setup was used to treat 5 mL of fresh distilled water from (RFD260NC, Advantec, Tokyo, Japan) a water distillation apparatus for 15, 30, and 45 min. The temperature of the electrode was confirmed, and ions and ozone generated were also measured. A UV-Vis spectrometer (V-670 Jasco, Pfungstadt, Germany) was used to analyze the absorbance of the reactive species dissolved in the distilled water after treatment.

### 3.3. Preparation of Bacterial Liquid

#### 3.3.1. Staphylococcus aureus

A bacterial strain of *S. aureus* NBRC 13276 obtained from the NITE biological resource center, Chiba, Japan, was used in this research; 1.3 g of Diago condensate powder 702 was diluted in 100 mL distilled water in a beaker then stirred until the mixture became homogeneous. The solution was transferred from the beaker into a glass bottle and heated in an autoclave (HRS-232, Hirayama, Saitama, Japan) at 121 °C for 30 min and allowed to cool at room temperature. After this process, 0.2 mL of liquid containing the bacteria was transferred with a syringe into a glass bottle containing 100 mL of liquid 702. This solution was incubated for 3 days at 37 °C.

#### 3.3.2. Cutibacterium acnes

*C. acnes* NBRC 107605 obtained from the NITE biological resource center, Chiba, Japan, was used in this research. To culture the bacteria used in this research, 1.77 g of Gam broth powder (05422, Nissui, Tokyo, Japan) was mixed with 30 mL of distilled water (RFD260NC, Advantec, Tokyo, Japan), then autoclaved (HRS-232, Hirayama, Saitama, Japan) at 121 °C for 30 min, then allowed to cool at room temperature. After this process, 0.2 mL of initial unknown bacterial concentration liquid was transferred with a syringe into a glass bottle containing 30 mL of Gam broth solution, then incubated for 3 days at 37 °C.

### 3.4. Samples Preparation

For *S. aureus*, mannitol Salt Agar 05236 was diluted with distilled water in a beaker (110 g of powder to 100 mL of distilled water). For *C. acne*, Gam Agar (05420, Nissui, Tokyo, Japan) was diluted with distilled water in a beaker (11.1 g of powder to 150 mL of distilled water) until the composition became homogeneous. These solutions were heated and sterilized in an autoclave at 121 and 115 °C, respectively, for 30 min. After cooling down, it was poured into petri dishes and left until the agar completely solidified. For the direct treatment, 50 µL of diluted bacteria at a concentration of 10^−5^ was spread onto agar in each petri dish, then treated with direct microplasma, while for the indirect treatment, 100 µL of PAW was added onto 50 µL of diluted bacteria at the same concentration. To show the bactericidal effect of both direct microplasma treatment and PAW on the samples, five (5) samples per condition were used. Finally, the samples were incubated at 37 °C for 48 h and 72 h for *S. aureus* and *C. acnes*, respectively.

## 4. Conclusions

We demonstrated the bactericidal efficiency of DBD microplasma through direct treatment and indirect bacterial inactivation through plasma activated water at low discharge voltages on *S. aureus and C. acne*. Under the experimental parameters used in this research, we conclude that direct microplasma treatment with air as a working gas, at a discharge voltage of 900 V and 27 kHz frequency, has a better bactericidal effect with 2 log reduction in CFU/mL (99%) at 3 min over PAW treated at 2.4 kV discharge voltage and 30 kHz frequency with 1 log (90%) reduction at 45 min. PAW treatment according to our setup generated less ozone concentration of 27.1 ppm at 1.2% O_2_ compared with 47.0 ppm for microplasma treatment using air. An increase in O_2_ percentage in gas mixture increased the discharge voltage in the indirect (plasma jet) treatment. An increase in treatment time caused a decrease in O_2_ concentration in the treated water at 2.4 kV using 1.2% O_2_ and 98.8% Ar, which also caused a decrease in the absorbance of the peaks at 210 nm, which are attributed NO_2_^−^ and NO_3_^−^ concentration in the treated water. The concentration of ozone almost doubled in the indirect treatment using plasma jet from 15.5 ppm at 0.8% O_2_ to 27.1 ppm at 1.2% O_2_, leading to the oxidation of NO_2_^−^. Finally, since the plasma jet’s bactericidal effects are limited to its surface area; activated medium such as PAW may increase the bacterial inactivation area larger than the electrode’s cross-sectional area, since a liquid medium could be spread on large surfaces.

## Figures and Tables

**Figure 1 molecules-26-02523-f001:**
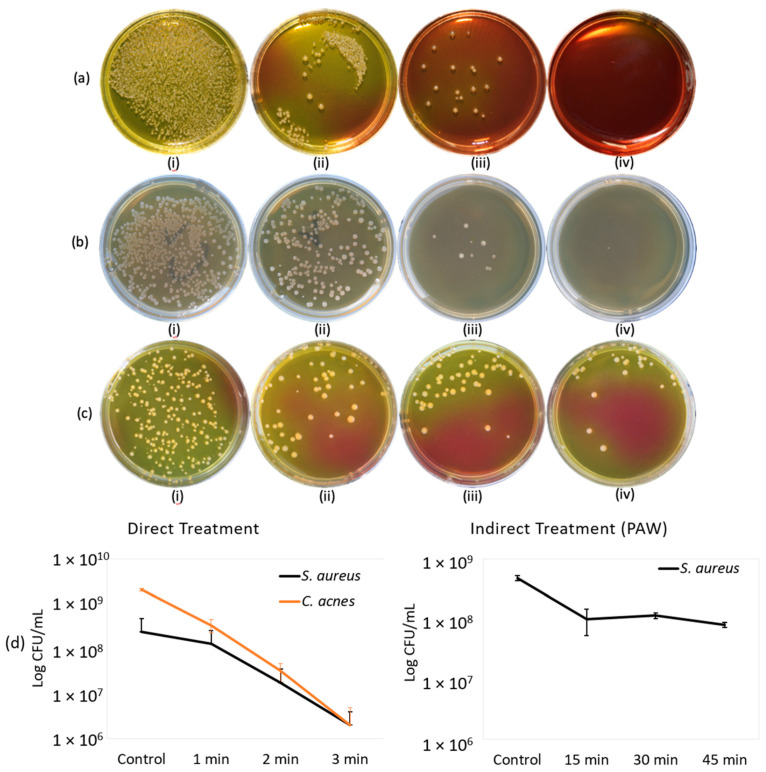
Bactericidal effect of direct microplasma treatment (**a**,**b**) and indirect treatment with PAW (**c**). (**a**) *S. aureus*; i. control, ii. 1 min, iii. 2 min, iv. 3 min. (**b**) *C. acnes*; i. control, ii. 1 min, iii. 2 min, iv. 3 min. (**c**) *S. aureus*; i. control, ii. 15 min, iii. 30 min, iv. 45 min. (**d**) Log reduction in survived bacterial colonies in CFU/mL for (left) direct treatment (right) and indirect treatment (PAW).

**Figure 2 molecules-26-02523-f002:**
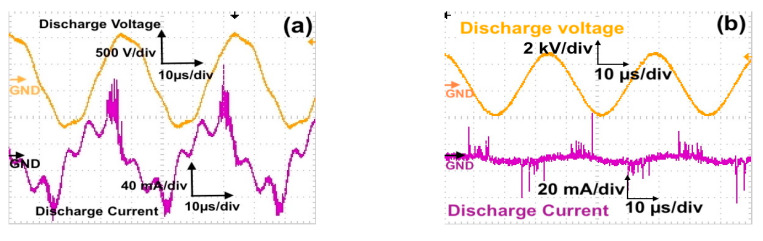
Discharge voltage and corresponding discharge current of (**a**) direct microplasma 900 V, air 2 L/min; (**b**) indirect treatment (plasma Jet) 2.4 kV, 1.2% O_2_ 3 L/min. Lissajours curve for (**c**) microplasma at 900 V, 27 kHz; (**d**) plasma jet at 2.4 kV, 30 kHz.

**Figure 3 molecules-26-02523-f003:**
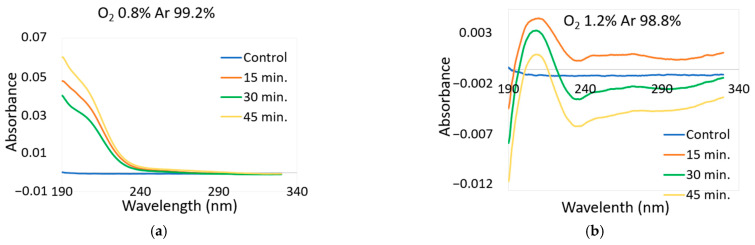
UV absorption spectra for PAW: (**a**) 0.8% O_2_ and 99.2% Ar; (**b**) 1.2% O_2_ and 98.8% Ar.

**Figure 4 molecules-26-02523-f004:**
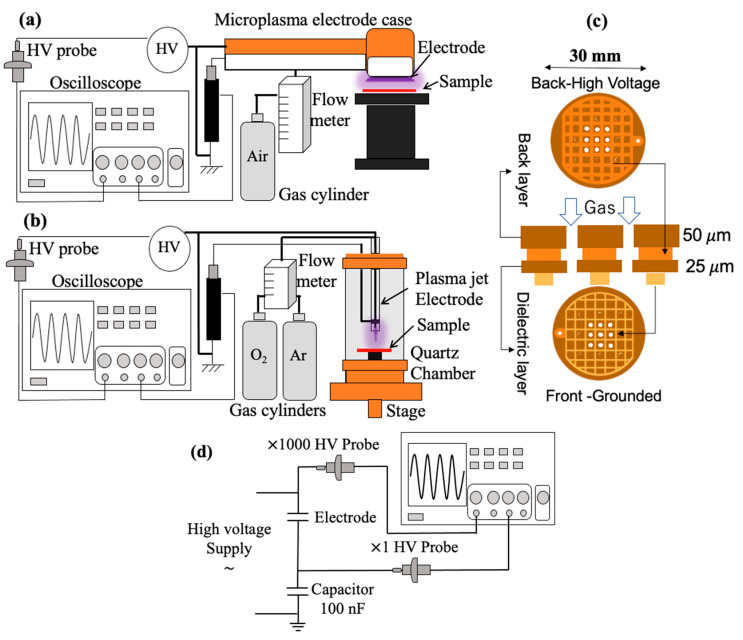
(**a**) DBD microplasma setup, (**b**) DBD microplasma electrode geometry, (**c**) plasma jet setup, and (**d**) transferred charge measurement setup.

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
