# Peer review of "Direct and Indirect Bactericidal Effects of Cold Atmospheric-Pressure Microplasma and Plasma Jet"

_molecules, 2021, doi:10.3390/molecules26092523_

Round 1

Reviewer 1 Report

The manuscript submitted by Yahaya et al. deals with the phenomenological study of the bactericidal effect produced by cold atmospheric-pressure micro-plasmas. The topic addressed in the work is in principal relevant for the Molecules journal. However, in the present state, the text requires extensive editing of English language, style and logical flow. Furthermore, in my opinion, this manuscript does not present results that might justify acceptance/publishing.

Comments:

  1. The authors should highlight/explain the novelty of this work. Is the novelty the fact that the RONS produce bactericidal effects? Is the novelty in the discharge setups? Is the novelty in the detections of species and/or analysis of processes?
  2. The Introduction is full of vague and unclear statements:                                                                                                  

    Page 1, ‘… conditions below 40°C’ … do you mean that the discharge at 45°C does not belong to the CAP category?

    P.1, ‘electrically generated highly energetic electrons’ … unclear statement

    P.1, ‘CAP has an advantage of being generated at low discharge voltage’ … do you mean that for example HV (50kV) pulsed corona at 20°C does not belong to the CAP category?

    P.2, ‘low discharge voltage bactericidal effects of direct and indirect sterilization properties of cold atmospheric pressure microplasma …’ … unclear statement

  3. P.2, ‘The number of (+ and -) ions generated at 900 V were 1.9 × 107 and 6.6×106 respectively.’ … meaning unclear. What ions and where? What’s the reason for (+)<<(-)? Non-neutral plasmas?
  4. P.3, ‘…the actual active time of the plasma jet streamers irradiation in …’ unclear statement
  5. P.4, negative absorbance in fig. 4b … any explanation?
  6. P.5, Presented discharge parameters and its characteristics are insufficient. I do not find even elementary ‘discharge’ information, such as the discharge power or energy density.
  7. P.5, ‘Increase in oxygen concentration in the gas mixture caused increase in absorbance of the peaks at 210 nm, this was attributed to increase in NO2− and 134 NO3− concentration in the water.’ … have you determined the PAW composition, at least approximately?
  8. P.5, ‘…each with a gap distance of 50 μm. The electrodes configuration was a grid shape gold wire with an interval of…’… the description of the DBD source is unclear (the electrode geometry)
  9. P.5, ‘… petri dish containing samples was set at 2 mm via a grounded micrometric sample holder’ ... how to exclude DBD between HV electrode and Petri dish in this case?
  10. P.6, concerning the indirect treatment: 5ml of water exposed to 3slm flow of a dry gas for 45 minutes … I would expect a complete evaporation or at least a significant reduction of the initial volume …?

Author Response

Thank you very much for your comments. According to the reviewer’s suggestions, we have answered and modified the manuscript as suggested.

Reviewer 2 Report

This manuscript attempts to investigate the bactericidal effects of direct microplasma treatment and plasma activated water (PAW) generated by a plasma jet. Although many articles have already discussed the bactericidal effects of either plasma or PAW, the authors combined these two treatment into one article. Unfortunately, by using two totally different plasma sources (plasma device, driving gas, power settings), the results could not provide systematic comparison of these two treatments, hence limiting the scientific impact of this manuscript. In addition, there is also only a very limited new insight regarding these two treatments. The pH decrease and gas phase ozone was investigated in multiple previous papers. Maybe the authors could investigate in depth the chemical properties in PAW. Therefore, I do not recommend to publish this manuscript in the current form.

Additional issues are listed below:

General points:

  1. Why are there two plasma sources used? For comparison, logically it would make sense to use the same source. Are there any particular reasons?
  2. The term “sterilization” in the results should be avoided since sterilization by definition means 6 log reduction.
  3. For the plasma jet different voltages for different gas mixture are used. Is there a particular reason? Additionally, it should be also clearly described which gas mixture is used in different experiments.
  4. References are not properly formulated. Sometimes the last names are abbreviated instead of the first name. For example, in [19].

Introduction:

  1. Page 1 Line 34: It is possibly more precise to say that CAP operates near or under room temperature rather than below 40 C.
  2. Page 1 Line 40: The term “discharge gap” is not clear. If the authors mean the gap between electrodes, I believe CAP sources do not have micrometer discharge gaps as well as 0.4-1.0 kV breakdown voltage.
  3. Page 2 Line 54: “Multiple papers…” This is true yet there are also many papers which uses lower voltage setups. It also strongly depends on the frequency and waveform.

Results:

  1. Page 2 Line 73: The effect of electric field should be neglected because it is relatively small. The discussion of the dielectric constant of water is not necessary since you either directly treat bacteria on agar dishes or with PAW.
  2. Page 2 Line 87: typo: 2which
  3. Page 2 Line 87: 9999% should be wrong here.
  4. Figure 1(b) i : For control, there are way too few bacteria. Is this a diluted sample? If yes, please mention it in your manuscript.
  5. Figure 2(b): for which gas mixture is this?
  6. Could you please explain why there were no PAW tests on C. acne?
  7. Figure 3: Could you please explain how you determined these results? You did not mention to use control samples with 10^8 – 10^9 CFU/ml. Is this a different experiment or the same as in Fig 1?
  8. Page 4 Line 121: typo: V-Vis Spectroscopy of Treated Water
  9. Page 5 Line 131: What is the physical meaning of the negative absorbance in your results here? And why does oxygen in water results in such outcome? There is also a negative value around 240 nm. Could you commend on this? Could there be an elimination of RONS? If yes, does it mean that there are less RONS after plasma treatment? Generally, Fig 4(b) is quite confusing. It seems that you adjusted the baseline of the results. An additional consideration point is that you have almost doubled the concentration of ozone in gas phase by comparing 1.2% O2 to 0.8% O2.

Materials and Methods:

  1. The description of the microdischarge setup is not easy to follow. I would recommend to add an illustration of the electrodes in detail in Fig. 5(a). Fig. 5(a) should also be drawn more clearly. Why is the HV connected to the microplasma probe and why is the probe above the electrode?
  2. Page 6 Line 164: Please provide more details regarding UV-Vis spectroscopy. For instance, about the apparatus etc.
  3. For PAW, one important factor is the storage time since it could affect the bactericidal effects. How long was the PAW stored before inactivation experiments? Or was it immediately used after plasma treatment?

Conclusion:

  1. Page 6 Line 199: S. aureus and C. acne should be in italic.
  2. Page 7 Line 206: “Increase in oxygen percentage…”, from which experiment could you conclude this?
  3. Page 7 Line 207: “Finally, plasma jet activated...” The meaning of this sentence is not clear.

Author Response

(The authors gave the same response as above.)

Round 2

Reviewer 1 Report

The authors have revised their original manuscript according to the comments made by both referees. However, I would suggest further modifications and improvements.

Comments to be considered:

  1. Editing of English language is required.
  2. Page 2, lines 45-47: This sounds like a general statement, which is however not correct. There are many DBD configurations requiring much higher voltages (tens of kV), still producing  microfilaments. Please rephrase/modify.
  3. Page 3, l.109-112: Still unclear how and where you sample +/-  ions … ? What’s the reason for (+)<<(-)? Non-neutral plasmas? References 40-41 are not relevant to your case. Please comment/justify/modify.
  4. Page 4, l.120-121: A streamer is definitely NOT a spark.  Please rephrase/modify.
  5. Page 5, l. 123-126: Current waveforms presented in figure 2 show, first off all, capacitive current. The discharge current peaks are of much shorter duration. Please comment/justify/modify.
  6. Clear description of the voltage/current/transferred charge measurement setup is completely missing and needs to be developed.   
  7. Page 8, setup figure and related text: according to part (b) of the figure, there is always free conductive path between HV and. grounded electrodes. Is this a DBD configuration ? Please comment/justify.

Author Response

Thank you very much for your kind comments and wonderful suggestions.

The manuscript's level of English was upgraded and modified according to reviewer's comments, see attached file please.

Reviewer 2 Report

In the revised version the authors corrected typing errors and some small mistakes. But their answer about the novelty of the article, which was using relatively low voltage discharges, is not convincing enough for me. Therefore, I recommend to either provide more information or at least to make some discussions.

First of all, reformulation is necessary. The structure of the manuscript is not easy to follow and sometimes misleading. Just to name a few, in subsection 2.2 there is a large amount of text explaining the electrical properties which should be in an independent section. Fig 1 and 3 are both showing inactivation results with similar conditions which could be combined into one figure (or at least put close to one another) for better reading. In section 3.2, the Lissajous measurements were done on both microplasma and jet. Putting the description there is misleading.

There are two conditions applied on the plasma jet (0.8% O2/2.2 kV and 1.2% O2/2.4 kV). These conditions were clearly stated in section 2.2 yet not in section 3.2 which is more important for the readers. Expect in the conclusion, it is also not clearly mentioned which condition was used for PAW bacteria inactivation.

Generally, the current form of this manuscript provides insufficient scientific impact for publication since inactivation of bacteria by either DBD plasma or PAW was shown in many previous publications even under lower voltage settings. However, the authors indicated some interesting result, for instance, the decrease of O2 and nitrite/nitrate in water with increase of treatment time. The manuscript will be significantly improved if the authors could provide additional data such as H2O2 or OH which was mentioned in the text. If not, a deeper discussion would also enhance the quality of the paper. Several question could be discussed including the decrease of O2 and the impact on nitrite/nitrate. The gaseous ozone is almost doubled under 1.2% O2 jet condition compared to 0.8%. Could this somehow impact the liquid phase RONS? Ozone has a relatively high reaction rate with nitrite, could this be a reason? Also, a comparison of the 0.8% and 1.2% O2 jet conditions on PAW bacteria inactivation could be interesting since they provide a very different plasma liquid chemistry.

Introduction:

  1. The authors briefly explained the microplasma. It would make sense if the authors also include a short introduction regarding plasma jets.

Results:

  1. Line 95: 2-log reduction is 99%
  2. Line 95: did you also do this measurement on the plasma jet?
  3. Line 96: The term “outside the microplasma” is not precise enough. I would suggest using terms like post-discharge region 3 cm below the grounded grid.
  4. Line 97: Units of ions are missing.
  5. Fig 2 (c)(d): please adjust the size, script, and font of these two images. Some texts are overlapping and some are missing.
  6. Line 126: Please add the gas and power setting for plasma jet bacteria treatment.

Materials and Methods:

  1. Figure 5(a): red line below the text “Microplasma electrode case”

Author Response

Thank you very much, we appreciate your kind efforts, comments, and contributions to the success of this paper. The manuscript was modified according to your suggestions/comments and attached belows

Round 3

Reviewer 1 Report

This revision brings the manuscript close to be acceptable. I would suggest only several minor amendments:

  • Page 4, last sentence ‘streamer discharge, also known as filamentary discharge’: The streamer is usually of filamentary nature at high pressures but certainly not all filamentary discharges are streamers. Consider rephrasing the statement.
  • Page 10, last paragraph ‘Tektronix (voltage ratio 1000:10)’ and ‘the transition charge’: Probably 1000:1 is correct? I would also suggest using ‘transferred charge’.
  • List of references, ref 41: I do not understand why the ‘stellator conditions’ should be relevant for the conditions of this study. Furthermore, text [41] in Russian language are hardly accessible to a broader audience. If this kind of reference is substantial for present study, it should be clearly explained in the text. Consider replacing with suitable reference which is relevant for atmospheric pressures (and non-fusion devices).

Author Response

Thank you very much for your kind comments and suggestions, the paper was modified according to reviewer's comments.

We really appreciate your efforts and contributions to this paper.
